# A 13-Year Approach to Understand the Effect of Prescribed Fires and Livestock Grazing on Soil Chemical Properties in Tivissa, NE Iberian Peninsula

**Meritxell Alcañiz [1], Xavier Úbeda [1] and Artemi Cerdà [2],*** 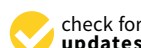

[1]    Grup de Recerca Ambiental Mediterrània (GRAM), Department of Geography, University of Barcelona, Montalegre 6, 08001 Barcelona, Spain; m.alcaniz@ub.com (M.A.); xubeda@ub.edu (X.Ú.)

[2]    Soil Erosion and Degradation Research Group, Department of Geography, Valencia University, Blasco Ibàñez, 28, 46010 Valencia, Spain

*    Correspondence: artemio.cerda@uv.es

**Abstract:** The high density of fuel accumulated in the Mediterranean ecosystems due to land abandonment results in high severity fires. Traditional fire practices and livestock grazing have played an important role in shaping the structure and composition of Mediterranean landscapes, and both can be efficient tools to manage them now that land abandonment is widespread. Attempts at controlling forest fires are essential for landscape management practices that, in their turn, seek to maintain a specific species composition. Against this backdrop, this study aims to determine the short- and long-term effects of the combined management practices of prescribed fires and goat grazing on the chemical properties of soils in Tivissa, Tarragona (NE Iberian Peninsula). Forty-two samples were collected in a $4 \times 18$ m plot before the prescribed fire of 2002 (1), immediately after the 2002 prescribed fire (PF) (2), one year after the 2002 PF (3), three years after the 2002 PF (4), and thirteen years after the 2002 PF (5). Soil samples were taken at each sampling point from the top layer (0–5 cm), sieved to obtain a <2 mm fraction, and soil pH, EC, Total C, total N, available P, $K^+$, $Ca^{2+}$, and $Mg^{2+}$ were determined. The results indicate that the short-term effects of fire are more relevant than those attributable to the livestock over the long term due to the low grazing intensity of less than one goat per ha. The long-term effects of prescribed fires were not visible in the research, suggesting that they recovered after burning with all their functions intact and with enhanced levels of natural fertility. Combined land management practices of prescribed fire and livestock grazing did not affect soil chemical properties. The applied management enhanced soil fertility and boosted the ecosystem's resilience.

**Keywords:** fire; ecology; nutrients; fertility; landscape; management; grazing; goats; plants; mountains

## 1. Introduction

Fire is a global phenomenon and a key ecological factor in the Earth's dynamics, especially in fire-prone ecosystems [1,2] such as that of the Mediterranean basin, where climate, flammable vegetation, and rugged terrain exacerbate the role of fire [3]. The Mediterranean has been exposed to the effects of fire during the Quaternary Period, modifying its landscapes and endowing many species with adaptive mechanisms that allow them to persist and regenerate after recurrent fires [4]. Thus, the Mediterranean ecosystems evolved with fire. Seen in this light, Mediterranean fires should not be considered as disasters, but rather as natural processes. However, today, forest fires (or wildfires) represent a major concern for many Mediterranean countries owing to the socio-economic changes such as the industrialization and rural population exodus in recent decades [3,4]. These

changes account for the current landscape pattern in which forests have replaced ancient agricultural fields, thereby increasing plant cover and the continuity of vegetation and, as a result, changing the fire regime and increasing wildfire risk [5]. Landscape changes are also attributable to fire suppression policies. They were introduced in response to the increase in wildfire activity since 1960 and based on the extinguishing of fires to reduce their spread. Prescribed fires were not allowed until recently. The objective was to remove fires from the forest, although forest fires are part of the natural Mediterranean ecosystems. These actions have facilitated the accumulation of fuel in Mediterranean forests, making them veritable tinderboxes waiting to burn [6,7]. Furthermore, fire affects the fate of soil infiltration, soil erosion processes and rates, vegetation changes, soil properties, and ecosystem services [8–12].

The results of years of research clearly point to the need to manage Mediterranean landscapes to minimize wildfire events and their impact [13]. It has been shown that fire management strategies can indeed reduce the fuel load of the region's forests, which, in themselves, constitute a valuable resource that generate many social benefits [14]. Prescribed fires (PFs) and prescribed grazing are two such strategies that allow these goals to be attained without them having dramatic ecological consequences for the environment. A prescribed fire is the planned use of fire under predetermined conditions (weather, fuel, and topography) and with clearly defined objectives [15,16]. This technique has become an increasingly viable alternative as a tool for fighting wildfires [17] insofar as it reduces fuel accumulation in Mediterranean forests and creates open spaces to facilitate the work of firefighters. However, prescribed fires can have other objectives, including the regeneration of certain plant species, the protection of habitats for mammals, and the grazing of shrublands in encroached lands based on a flexible, low-cost technique [18,19]. Figure 1 shows examples where fire is present and takes an important role in ecosystem evolution. Fire is a planet process that shapes Earth landforms, soils, biota, and atmosphere composition.

However, in recent decades, grazing has generally been perceived as negative due to problems of overgrazing and the use of fire for pasture renewal [20–22]. Yet, in Mediterranean areas, undergrazing is in fact more of a problem, given that the heavy grazing of woody vegetation in the region constitutes one of the most efficient management techniques for preventing fire and maintaining habitat diversity [23]. At the same time, it is also arguably the most competent technique for eliminating the vertical and horizontal continuity of fuels, while preserving species diversity and reducing wildfire risk [22]. The goat is the most suitable species for this purpose because of its browsing ability, which means that it can forage for resources that other ungulates, such as sheep or cattle, cannot [24]. A goat's annual diet is highly varied, but it is made up on average of about 60% shrubs, 30% grasses, and 10% forbs [25]. As such, goats browse a greater variety of plants than other classes of livestock and so help restore the cycling of nutrients sequestered by woody species [26]. Moderate levels of grazing have been shown to modify wildfire behavior, slowing their spread, shortening flame length, and reducing fire intensity [27]. At the same time, goats preferentially consume seeding stems and so reduce the spread and perpetuation of weeds by seed [26]. However, the most sustainable strategy would appear to involve the integration of different treatments: thus, one solution might be the use of prescribed fires to eliminate large diameter vegetation while grazing can be employed as a follow-up treatment to control resprouting species or to shift the species composition to herbaceous plant fuel material [28]. Grazing is much more than a simple biological or economic process; it is also a sociopolitical activity in that it involves various sectors of the community and the attitudes they hold to the landscape [23]. Other factors to take into account in this research are as follows: (i) the properties of the fire (intensity, residence time, and seasonality) [29,30]; and, (ii) the properties of the prescribed grazing: livestock species, grazing experience, plant physiology [nutrient content], stocking density, grazing duration, and animal's physiological state [28]. Prescribed fires and prescribed grazing can either be a short-term measure to reduce flammable vegetation or a long-term measure to modify vegetation composition by depleting root carbohydrates in perennials and by reducing the soil seed bank for annual plants, in an effort to change fire behavior, fuel loading, plant cover, and ladder fuels [22]. The impact of

such measures depends on the type of vegetation and the phenological state of the plants, the season, the frequency and intensity of burning, grazing pressure, the physical structure of landscape, and the climatic conditions [31,32].

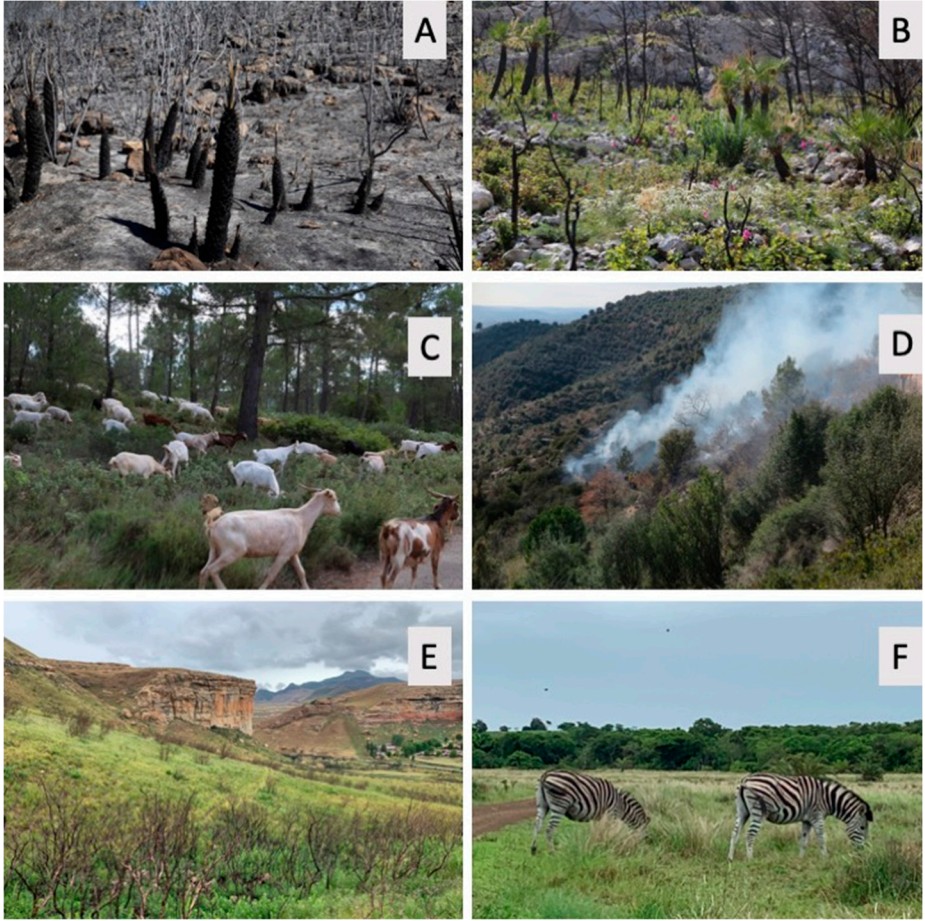

**Figure 1.** Fire and grazing shapes the landscape of the world. (**A**), Recently burnt (2 days) Mediterranean macchia in Carcaixent (València province, Eastern Iberian Peninsula) June 2016. (**B**), Vegetation recovery after the fire of Cala de la Granadella (Alacant province, Eastern Iberian Peninsula) April 2017 (wildfire August 2016). (**C**), The use of goats to reduce fuel at the Bicorp (Massís del Caroig, inland Valencia Province, Eastern Iberian Peninsula). (**D**), Prescribed fire in Olesa de Montserrat, Catalonia, Janauary 2017. (**E**), View of the recently burnt area by shepherds at the Golden Gate Highlands National Park, Free State, South Africa (December 2019). (**F**), Natural grazing act as a vegetation biomass control. Zebras grazing in Hluhluwe Infolosi Park, KwaZulu Natal South Africa (December 2019). Photographs by Artemi Cerdà.

The interaction of fire and grazing on the ecosystem's evolution has been researched widely around the world [33–35]. However, the investigations are very little in the Mediterranean ecosystems, and no information about how prescribed fires affect the recovery of the ecosystems when grazing is present. Prescribed forest fires are widely used today around the world, and they are a successful strategy to maintain the fire as a natural ecosystem factor, avoid the damage and risk of the wildfires, preserve the soil nutrient availability, and reduce the soil erosive potential [36–38]. Wildfires used to cause deep changes in the vegetation, soil properties, soil, and water yield as a consequence of sudden floods and risk of land degradation [39–42].

The study reported here is innovative insofar as little research to date has examined the combined effects of prescribed firesand livestock grazing on soil chemical properties. As such, the main

purpose of this study was to quantify short- and long-term changes in soil properties in a regime of recurrent prescribed fires and goat grazing in a Mediterranean shrubland. More specifically, the study sought to observe the evolution in soil chemical properties in a 13-year period since the first fire management intervention.

## 2. Materials and Methods

### 2.1. Study Area

The study area lies in the Tivissa mountain range in the province of Tarragona (NE Iberian Peninsula) (Figure 2) at an altitude of 615 m.a.s.l. The plot slope is ca. 35% and the soil is classified as a Lithic Calcixerepts, with a calcareous bedrock. The area has a Mediterranean climate with a mean annual precipitation of 581 mm and a mean temperature of 15.7 °C [30]. The plot was occupied by an almond plantation until 1989, the year in which a wildfire transformed the landscape. After this event, the plantation was abandoned, and the area was colonized by herbs and shrubs, increasing the risk of wildfire because of the associated accumulation of fuel. Before the first prescribed fire, the vegetation was dominated by shrubs, occupying ca. 75% of the area: *Ulex parviflorus* Pourr. (60%), *Cistus albidus* L. (30%), *Rosmarinus officinalis* L.; *Erica multiflora* L.; and *Quercus coccifera* L. The plot's herbaceous vegetation was composed primarily of *Brachypodium retusum* P. Beauv. A tree stratum—represented by *Pinus halepensis* Mill.—was also present but occupying just 1% of the plot [30].

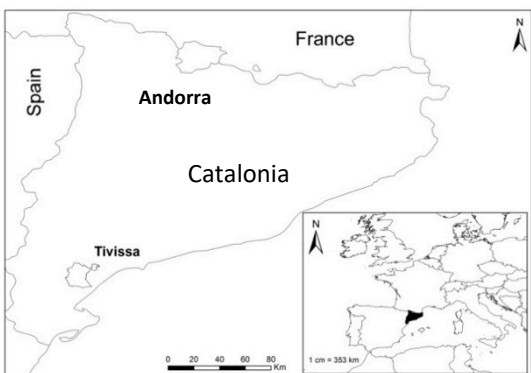

**Figure 2.** Location of the study area of Tivissa in the south of Catalonia.

The first prescribed fire was set by the GRAF unit (Forest Action Support Group—forming part of the Catalan Fire Department) in May 2002, while the second was set nine years later in 2011. In relation to the first event, the GRAF unit identified two main forest management objectives: (1) to make *Cistus albidus* L. the dominant shrub species (where previously it had been *Ulex parviflorus* Pourr.) so as to have a less flammable vegetation structure and (2) to permit the introduction of livestock into the area. A few months after the first prescribed fire, a herd of goats were introduced on these slopes, and today, 300 goats graze there each winter with a density of less than 1 goat per ha. As such, the objective of the subsequent application of fire in this plot was more closely related to plant regrowth and palatability. The first prescribed fire (2002) was classified as being of medium severity, and the second (2011) was classified as being of low severity by the unit [30].

### 2.2. Methods

#### 2.2.1. Soil Sampling and Laboratory Analysis

Soil samples were collected on five sampling dates: (1) before the 2002 prescribed fire (PF), (2) immediately after the 2002 PF, (3) one year after the 2002 PF, (4) three years after the 2002 PF, and (5) thirteen years after the 2002 PF (and four years after the 2011 PF). The sampling design used in

this study is the same as that described for the Montgrí plot [30], comprising 42 sampling points within a plot, measuring 4 × 18 m, within the burned area. Figure 3 show the view of the study site in 2006 and 2015. A sampling plot was established within the burned area using a grid structure (4 × 18 m). Soil samples were collected after the prescribed fire at 42 points along three transects and along three lines running perpendicular to the central transect [30].

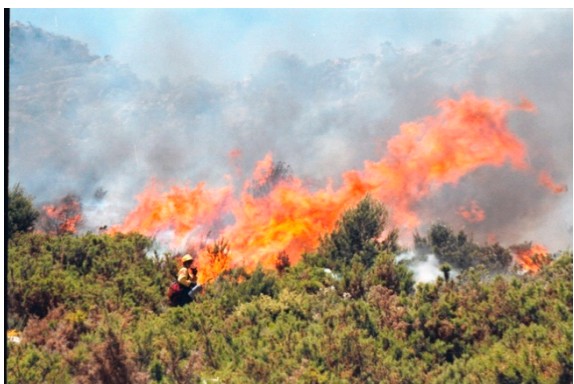 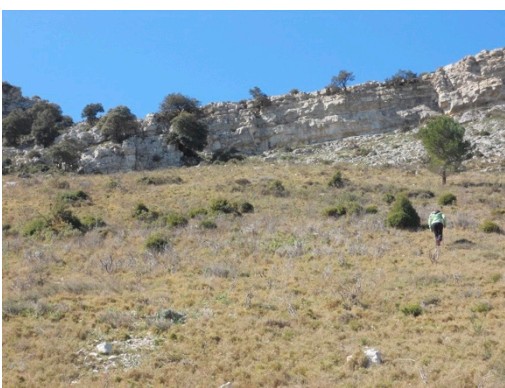

**Figure 3.** Tivissa plot during the first prescribed fire in 2006 (to the **left**, by Xavier Úbeda) and as seen nine years later during the most recent sampling event (to the **right**, by Meritxell Alcañiz).

Soil samples were taken at each sampling point from the top layer (0–5 cm) using a small pick. Then, ash was removed, and the sample was air dried in the laboratory and sieved to obtain a <2 mm fraction. Following extraction with deionized water, soil pH (1:2.5) and electrical conductivity (1:2.5) were analyzed and measured with a pH meter and a conductometer, respectively [31]. Total C and total N were determined using elemental analysis (NaA2100 Protein Nitrogen Analyzer, Usmate Velate (MB) Italy. Available P was analyzed using the Olsen Gray method [32]. Exchangeable cations, $K^+$, $Ca^{2+}$, and $Mg^{2+}$ were analyzed by ammonium acetate extraction [33] and determined by atomic absorption spectrophotometry.

### 2.2.2. Data Treatment

Descriptive statistics were calculated, and the Kolmogorov–Smirnov test was used to test the assumption of normality at a 95% confidence level. Some variables satisfied the null hypothesis, so a one-way ANOVA test was conducted to identify statistically significant differences between different sampling dates at the 95% confidence level. However, some data were not normally distributed and, so, the Kruskal–Wallis test was used to identify statistically significant differences (95%) at different sampling dates, namely prior to the prescribed fire, immediately after the prescribed fire, and then one year, three years, and thirteen years after the event.

### 3. Results

The results obtained from the five sampling dates are shown in Table 1. *p*-values from the one-way ANOVA test are shown in Table 2 and indicate whether the differences between sampling dates are significant ($p < 0.005$) or not ($p > 0.005$).

Immediately after the first PF, pH values increased significantly from 7.61 ± 0.01 to 7.74 ± 0.02 (Tables 1 and 2). One year after the PF, pH values had risen even higher (8.13 ± 0.03) but had begun to fall three years after this PF. Yet, values had not returned to their pre-fire level (7.75 ± 0.03) thirteen years after the first PF and three years after the second PF set in the same plot. Likewise, mean EC increased significantly immediately after the PF, rising from 326.52 ± 10.34 μS cm$^{-1}$ to 439.46 ± 20.67 μS/cm (Tables 1 and 2). A year later, EC values had fallen but had not yet reached their pre-fire levels (378.36 ± 12.90 μS cm$^{-1}$).

**Table 1.** Minimum, maximum, mean (bold), standard deviation (SD), variance, and standard error (SE) values before the prescribed fire (PF), after PF, one year after PF, 3 years after PF, and 13 years after PF.

| Soil Properties | Statistics | Before PF | After PF 0 Year | After PF 1 Year | After PF 3 Years | After PF 13 Years |
|---|---|---|---|---|---|---|
| pH | Min | 7.42 | 7.40 | 7.55 | 7.38 | 7.20 |
| | Max | 7.75 | 7.95 | 8.46 | 8.13 | 7.97 |
| | **Mean** | **7.61** | **7.74** | **8.13** | **7.81** | **7.75** |
| | SD | 0.09 | 0.10 | 0.22 | 0.13 | 0.16 |
| | Variance | 0.01 | 0.01 | 0.05 | 0.02 | 0.02 |
| | SE | 0.01 | 0.02 | 0.03 | 0.02 | 0.03 |
| EC ($\mu$S cm$^{-1}$) | Min | 238.00 | 260.00 | 269.00 | 159.90 | 116.60 |
| | Max | 535.00 | 801.00 | 688.00 | 300.00 | 204.50 |
| | **Mean** | **326.52** | **439.46** | **378.36** | **216.20** | **167.89** |
| | SD | 66.99 | 133.95 | 83.57 | 33.23 | 24.70 |
| | Variance | 4487.18 | 17,942.55 | 6984.58 | 1104.31 | 610.12 |
| | SE | 10.34 | 20.67 | 12.90 | 5.13 | 4.51 |
| EC (%) | Min | 8.22 | 8.43 | 6.77 | 6.53 | 6.55 |
| | Max | 15.65 | 23.82 | 19.00 | 16.30 | 14.24 |
| | **Mean** | **10.63** | **12.19** | **10.36** | **9.70** | **8.78** |
| | SD | 1.89 | 3.21 | 2.41 | 2.19 | 1.39 |
| | Variance | 3.57 | 10.31 | 5.79 | 4.79 | 1.95 |
| | SE | 0.29 | 0.50 | 0.37 | 0.34 | 0.25 |
| SOC (%) | Min | 5.21 | 6.06 | 2.98 | 0.70 | 0.06 |
| | Max | 18.03 | 26.57 | 17.55 | 13.88 | 16.82 |
| | **Mean** | **9.34** | **11.80** | **5.97** | **5.69** | **5.37** |
| | SD | 2.83 | 4.01 | 2.35 | 2.94 | 3.79 |
| | Variance | 8.02 | 16.08 | 5.50 | 8.64 | 14.33 |
| | SE | 0.44 | 0.62 | 0.36 | 0.45 | 0.58 |
| TN (%) | Min | 0.39 | 0.44 | 0.43 | 0.37 | 0.40 |
| | Max | 1.09 | 1.55 | 1.12 | 1.19 | 1.11 |
| | **Mean** | **0.60** | **0.69** | **0.63** | **0.64** | **0.58** |
| | SD | 0.16 | 0.24 | 0.17 | 0.18 | 0.13 |
| | Variance | 0.03 | 0.06 | 0.03 | 0.03 | 0.02 |
| | SE | 0.02 | 0.04 | 0.03 | 0.03 | 0.02 |
| Ca$^{2+}$ (ppm) | Min | 8914.00 | 7881.00 | 9778.00 | 16,316.63 | 13,986.00 |
| | Max | 12,200.00 | 14,960.00 | 17,390.00 | 36,058.88 | 23,142.00 |
| | **Mean** | **10,389.32** | **9829.29** | **11,482.50** | **26,226.34** | **17,091.73** |
| | SD | 766.44 | 1197.69 | 1460.54 | 3790.68 | 2118.29 |
| | Variance | 587,429.17 | 1,434,471.04 | 2,133,250.01 | 14,369,256.38 | 4,487,132.06 |
| | SE | 118.26 | 184.81 | 225.37 | 584.91 | 386.74 |
| Mg$^{2+}$ (ppm) | Min | 4.20 | 147.60 | 117.20 | 233.77 | 185.68 |
| | Max | 370.10 | 525.90 | 981.50 | 430.32 | 400.21 |
| | **Mean** | **207.15** | **292.93** | **209.95** | **327.54** | **262.07** |
| | SD | 64.21 | 93.00 | 129.17 | 46.89 | 50.20 |
| | Variance | 4122.76 | 8649.87 | 16,685.91 | 2198.29 | 2520.42 |
| | SE | 9.91 | 14.35 | 19.93 | 7.23 | 9.17 |
| K$^{+}$ (ppm) | Min | 356.90 | 441.60 | 272.30 | 314.19 | 419.40 |
| | Max | 2203.00 | 3270.00 | 1861.00 | 908.53 | 784.37 |
| | **Mean** | **995.39** | **979.92** | **712.47** | **515.13** | **557.91** |
| | SD | 453.43 | 480.80 | 274.33 | 100.36 | 89.70 |
| | Variance | 205,596.05 | 231,169.61 | 75,258.18 | 10,073.03 | 8045.90 |
| | SE | 69.97 | 74.19 | 42.33 | 159.90 | 16.38 |
| P (ppm) | Min | 6.20 | 0.00 | 11.00 | 37.80 | 56.70 |
| | Max | 144.90 | 481.00 | 102.00 | 137.46 | 137.46 |
| | **Mean** | **60.91** | **116.54** | **45.77** | **77.84** | **77.98** |
| | SD | 32.12 | 104.85 | 21.28 | 19.43 | 18.12 |
| | Variance | 1031.94 | 10,994.44 | 452.99 | 377.72 | 328.31 |
| | SE | 4.96 | 16.18 | 3.28 | 3.00 | 3.36 |

**Table 2.** One-way ANOVA *p* values. Significant differences in bold. ($p < 0.005$; $n = 42$).

|  | pH | EC | TC | TN | Ca$^{2+}$ | Mg$^{2+}$ | K$^+$ | P |
|---|---|---|---|---|---|---|---|---|
| Before PF–After PF | 0.000 | 0.000 | 0.008 | **0.040** | 0.001 | 0.000 | **0.950** | 0.003 |
| Before PF–1 year | 0.000 | 0.001 | **0.289** | **0.452** | 0.000 | **0.900** | 0.003 | 0.016 |
| Before PF–3 years | 0.000 | 0.000 | 0.039 | **0.335** | 0.000 | 0.000 | 0.000 | 0.001 |
| Before PF–13 years | 0.000 | 0.000 | 0.000 | **0.506** | 0.000 | 0.000 | 0.000 | 0.012 |
| After PF–1 year | 0.000 | 0.004 | 0.001 | **0.161** | 0.000 | 0.000 | 0.000 | 0.000 |
| After PF–3 years | 0.001 | 0.000 | 0.000 | **0.227** | 0.000 | 0.023 | 0.000 | **0.904** |
| After PF–13 years | **0.138** | 0.000 | 0.000 | **0.180** | 0.000 | **0.178** | 0.000 | **0.945** |
| 1 year–3 years | 0.000 | 0.000 | **0.198** | **0.828** | 0.000 | 0.000 | 0.000 | 0.000 |
| 1 year–13 years | 0.000 | 0.000 | 0.001 | **0.179** | 0.000 | 0.000 | 0.001 | 0.000 |
| 3 years–13 years | 0.000 | 0.000 | **0.048** | **0.126** | 0.000 | 0.000 | **0.043** | **0.444** |

During the thirteen-year experimental period, EC has tended to fall, recording relatively lower values than those noted before the PF (for example 13 years after the PF, EC stood at 167.89 ± 4.51 μS cm$^{-1}$). Total C values also increased significantly in the immediate aftermath of the first PF (10.63% ± 0.29% vs. 12.19% ± 0.50%) but had fallen to a level below pre-fire values one year after (10.36% ± 0.37%) (Tables 1 and 2). Total Carbon (TC) continued to fall three years after the PF (9.70% ± 0.34%) and recorded even lower levels after the second PF (8.78% ± 0.25%). Similar to C levels, total N increased significantly just after the PF, rising from 0.60% ± 0.02% to 0.69% ± 0.04%, but had fallen to 0.63% ± 0.03% one year after the first PF. A relevant issue here is that the inorganic carbon estimated from the total carbon and the soil organic carbon measured increased about 4-fold in the post fire (one-year) period and continues throughout the 13-year time period. This is confirmed with the changes in the pH. This is a clear and significant change in soil chemical properties that results in a different plant composition and runoff generation and soil losses.

Thirteen years after the first PF and four years after the second treatment, total N values were not significantly different from pre-fire values (0.58% ± 0.02%) (Tables 1 and 2). Available phosphorous also increased after the PF, reaching levels that were almost twice those of the pre-fire values (from 60.91 ± 4.96 ppm to 116.54 ± 16.18 ppm). However, one year later, the values had fallen below pre-fire levels (45.77 ± 3.28 ppm) but thereafter rose significantly (77.84 ± 3.00 ppm three years after and 77.98 ± 3.36 ppm thirteen years after) (Tables 1 and 2).

The behavior of the inorganic cations varied (Tables 1 and 2); thus, while calcium levels fell in the aftermath of the PF, potassium levels presented no significant changes and magnesium levels increased significantly. Calcium levels continued to increase after the PF but fell after the second PF (last sampling date). One year after the PF, magnesium values had returned to their pre-fire values but subsequently rose to peak three years after the event (327.54 ± 7.23 ppm).

In more recent samplings, EC has tended to fall, recording relatively lower values than those noted before the PF (for example 13 years after the PF, EC stood at 167.89 ± 4.51 μS/cm). Total C values also increased significantly in the immediate aftermath of the first PF (10.63% ± 0.29% vs. 12.19% ± 0.50%) but had fallen to a level below pre-fire values one year after (10.36% ± 0.37%) (Tables 1 and 2). TC continued to fall three years after the PF (9.70% ± 0.34%) and recorded even lower levels after the second PF (8.78% ± 0.25%). Similar to C levels, total N increased significantly just after the PF, rising from 0.60% ± 0.02% to 0.69% ± 0.04%, but it had fallen to 0.63% ± 0.03% one year after the first PF.

## 4. Discussion

Previous studies have shown that the effects of burning depend on several factors including soil type, vegetation, topography, and a number of fire characteristics, namely, intensity, severity, residence time, seasonality, and period between PFs. Fire effects on ecosystem functioning is complex and multifaceted. Grazing is also a key factor to understand the Mediterranean ecosystem evolution. In this study, prescribed fires were combined with grazing to reduce vegetation density and achieve a specific vegetation structure that limits fire spread. Consequently, the use of prescribed fires and grazing

reduces the risks to wildfire. Prescribed or targeted grazing can be defined as the application of a specific kind of livestock during a given season, for a given duration and intensity to meet pre-defined vegetation/landscape goals [34]. Here, 300 goats are allowed to graze each year during the winter, providing the animals with food while, at the same time, managing the shrub vegetation.

The sampling conducted before and immediately after the PF showed statistically significant higher values in almost all parameters, with the exception of calcium levels, which fell, and potassium levels that underwent no significant change (Tables 1 and 2). However, the third sampling (one year after the event) typically showed these values to have fallen. Three years after setting the first PF, some chemical properties (pH, EC, and potassium) continued to fall, while others presented no significant changes. However, the inorganic cations (calcium and magnesium) both recorded maximum levels three years after the first burning treatment, while available phosphorous levels also increased but peaked just one year after the PF. The most recent results (i.e., thirteen years after the first PF and four after the second) show that the chemical parameters analyzed behaved quite differently; thus, while some (EC, total C, and total N) had returned to their pre-fire levels by this date (or even before), others continued to record values higher than their pre-fire levels. Despite documenting all these chemical changes in the soil, this research still needs to determine if it has been able to preserve all its primary functions, which is supporting, regulating, and provisioning the suite of ecosystem services [43].

## 4.1. pH and Electrical Conductivity

A good many studies conducted in different ecosystems around the world show that soil pH increases immediately after a PF [29]. This is attributable primarily to the incorporation of ash into the soil [35,36]; however, the oxidation of organic matter and the subsequent release of cations can also help raise pH values [37,38]. Here, soil pH reached its highest level one year after the PF, which was due in all probability to the leaching of ash into the soil in the months following the episode. Researchers observed similar behavior in the Montgrí plot [30], but thereafter, pH levels started to fall (Table 1). However, soil pH values are not always affected; for example, Alcañiz et al. [30] report that in roughly 50% of the studies they reviewed, there was no statistically significant change in pH.

Electrical conductivity is often found to increase after a PF [30]. In the Tivissa (henceforth TVA) plot, the EC rose after the PF, which might be due to the release of soluble inorganic cations, the incorporation of ash into the soil, and the formation of black carbon [39]. Thereafter, EC levels fell significantly at each sampling date (Table 1). This might be a result of post-fire surface runoff, especially given the slope of the plot [40]. Thirteen years after the PF, EC values are at their lowest recorded level, which is due probably to nutrient export to the shrubs that have been grown to feed the goats.

pH and EC levels do not seem to have been directly affected by grazing. However, while researchers observed the effects of the first PF set in 2002, those of the second set nine years later cannot be detected. However, the main reason for this is that sampling was not carried out until four years after the second PF—if sampling had been conducted before and after this event (immediately or after a year), then some differences were observed, given that these changes in soil chemical properties seem to disappear somewhere between the first and third years post fire (Table 1).

## 4.2. Carbon (C) Stocks and Nitrogen (N)

Table 1 shows that the behavior of total C, SOC (Soil Organic Carbon), and total N was largely similar: their levels increased in the aftermath of the first PF but thereafter they fell, recording their lowest values at the final sampling date. The effects of a fire on soil properties depend essentially on its intensity and severity [41]; thus, following low-intensity PFs, it was expected to find higher C values due to the incorporation of unburned, or partially combusted, material into the soil [30,41,42,44–47]. However, Alcañiz et al. [30] conclude that the aftermath a PF often has a neutral impact on C pools (63% of the studies reviewed therein) and, in some cases (13%), C depletion was actually recorded due

primarily, it would seem, to recurring fires. Indeed, Muqaddas et al. [43] report that fire recurrence plays an important role in the management of vegetation. In their study, they observed C losses when the PFs were set every two years, while no changes were detected when set every four years. In this case, there was a nine-year period between the fires, and similar effects were not detected. In short, a higher frequency of the application of fire appears to have consequences for these soil parameters.

As discussed, N levels behaved quite similarly to those of soil C. TN increased immediately after the PF, but thereafter, values returned to their pre-fire levels and even lower (Table 1). Alcañiz et al. [48] reported that in 88% of the studies reviewed, N stocks after a PF either increased or presented no significant changes. Normally in low-intensity PFs, that is, in which maximum temperatures do not exceed 200 °C and whose duration is not long [45], N stocks rise due to the incorporation of ash into the soil and the decomposition of the forest floor [43,46–48]. Yet, it seems that the effects of fire on soil N are quite ephemeral, because the one-year sampling values are not statistically significantly different from their pre-fire values. Equally, there is no overall loss of soil N because of the long nine-year period between events. Arguably, if burning was more frequent, the impact on soil N would be greater because of the volatilization of this element; for example, the studies of Scharenbroch et al. [47] and Muqaddas et al. [43] conclude that while biannual burning decreases N pools, burning every four years increased soil's N content.

Few studies have examined the combined effects on soil parameters resulting from PFs and livestock grazing [48]. Exceptions include Hiernaux et al. [49], who reported a fall in C content due to grazing pressure on soils; however, in the study area, C stocks increased just after the PF, which perhaps was a result of organic matter being returned to the soil in the goat feces and urine. However, Savadogo et al. [50], in a study of the effects of grazing intensity and a PF on a savanna woodland soil, found that soil chemical characteristics were affected—primarily C and N stocks—due to the intensity of grazing, and that these effects were especially notable on unburnt subplots. Ten years later, the authors reported that after two decades of annual early burning and moderate grazing, no significant effects were detected on soil C and N [51]. Taking into account the results reported, herein, it is concluded that the effects of goat grazing were not readily observable.

*4.3. Nutrient Availability*

The nutrient availability results show that the mean values of the parameters analyzed in the last sampling period are suitable for ensuring plant regrowth, these nutrients being essential for plant nutrition (Table 3) [52]. Each of the parameters presents its own distinct pattern of behavior: thus, while $Mg^{2+}$ and available P increased after the PF, $Ca^{2+}$ decreased and $K^+$ did not undergo any significant change. After a PF, nutrient contents typically increase due to (1) the release of basic cations from the organic matter; (2) ash formation and incorporation into the soil; (3) the high volatilization temperatures of the nutrients; (4) and, in the case of available phosphorous, the mineralization of organic phosphorous at high temperatures [30,45,52–54]. However, in response to increased pressure in their study pastures, Arevalo et al. [55] did not find any changes in soil calcium, magnesium, and potassium. Here, though, one year after the PF, nutrient content levels fell below post-fire values, but at the next sampling date, nutrient levels had risen again (i.e., before the second PF). These changes may reflect the grazing intensity in the plot in the three years after the first PF. Indeed, the higher nutrient values might be attributed to the deposition of urine and manure. However, it seems that the combined effect of livestock grazing and PFs is greatest in the immediate aftermath of the PF. However, note that Boughton et al. [56] reported the most significant changes in nutrient levels in association with pasture intensity, followed by the effects of PFs. In this study, the impact of livestock grazing was not evident until the fourth sampling data, probably reflecting the low intensity of the grazing (a herd of just 300 goats). Other studies identify the strong synergistic effects of PFs and livestock grazing on terrestrial pastures [57,58]. Moreover, the changes detected in soil nutrient availability are clearly going to shift vegetation composition and pasture structure [59–63]. Therefore, further studies examining the floristic and herb composition could shed light on the way in which nutrients are able

to change the vegetation and improve the palatable species favored by livestock, while at the same time, management plans are designed to reduce wildfire risk.

**Table 3.** Mean chemical property values recorded at the Tivissa (TVA) plot on the last sampling date (13 years after first PF) compared with values reported in soil reports for the area. B, Basic; H, High; M, Medium; L, Low; U, Above optimal).

| | pH | EC $\mu S\ cm^{-1}$ | TC% | SOC% | TN% | $Ca^{2+}$ ppm | $Mg^{2+}$ ppm | $K^+$ ppm | Ava P ppm |
|---|---|---|---|---|---|---|---|---|---|
| **Tivissa 13y** | 7.75 | 167.89 | 8.78 | 5.37 | 0.58 | 17091.73 | 262.07 | 557.91 | 77.98 |
| [64] | >6.5 B | - | - | - | - | >3000 H | >245 H | >25 H | >30 H |
| [65] | >7 B | <500 L | - | >2 H | - | >400 H | >30 H | >100 U | >50 U |
| [63] | - | 100–450 L | - | - | - | >2000 H | >180 H | 280–800 H | 40–100 H |
| [66] | 7–8 M | 100–400 L | - | >3.5 H | >0.2 H | >4000 H | <300 L | >300 H | >30 H |
| [67] | – | - | - | - | - | >300 H | >100 H | >80 H | >8 H |
| [68] | 7–8 M | - | - | 4–10 M | >0.5 H | >2000 H | 60–500 M | >250 H | >15 H |
| [62] | 7.5–8.5 B | <400 L | - | - | - | - | - | - | - |

### 4.4. Soil Quality after Fire and Livestock Grazing

We reviewed a number of publications/reports that provide information about the optimum ranges of different soil properties in this area in order to determine if soil conditions remained good on the plot after recurrent PFs and after being exposed to livestock grazing (Table 3). Note that in our study sites after the fire and grazing for thirteen years, all the parameters analyzed lie within the optimum ranges indicative of good soil quality in the TVA plot following the disturbances. In all the reports, pH levels are classified as basic or alkaline (>7) due to the calcareous bedrock in this area [54–76]. The low EC values—always below 400 μS/cm, which is roughly the highest value identified in the reports for a good soil—indicate that the soil is not saline [54–60]. The SOC level is higher than the limits reported, which can be attributed to the fact that the soil classifications correspond in the main to agricultural soils, which normally contain less organic matter. In this case, TVA is a natural prairie, which means there is no fertilization management and SOC inputs result essentially from the deposition of organic materials (vegetation, manure, etc.). Nitrogen content at the last sampling date is also above the limits identified by the classification [60] as ensuring the correct functioning of the soil ecosystem. However, nutrient availability to ensure vegetation recovery after the PF or after grazing is guaranteed because of the high levels of all the other parameters analyzed. In conclusion, and after comparing the results with the levels reported in the literature, it is confirmed that the soils in this plot have good nutrient, salinity, and acidity levels to perform their main functions; that is, they support both animals and plants. The soil system sustains the main ecosystem services in the Earth system [61], and this is relevant to achieve the Sustainable Development Goals of the United Nations [62] and the Land Degradation Neutrality Challenge [63,65], which will also affect the fire-affected land around the world and where we need to find proper management. Here, we found that prescribed fires with grazing are a sustainable solution to the management of fire-affected land, which is under debate [64,66,67]. Other researchers found that after wildfire, the recovery of the vegetation and the soil properties induced high infiltration rates [68]. Forest fires are accused of affecting the soil organic matter content [69] and then the atmospheric composition; however, their services as part of the ecosystem are relevant [70]. Previous research found that vegetation recovery after a long period (18 years) was very positive in a Catalonia location, too [71]. Trail erosion in recently burnt areas is a threat [72], but there are many strategies to make the immediate post-fire period sustainable with the use of straw mulches [73], which is a strategy imported from agriculture land where the problem of erosion and soil degradation is a persistent problem [74]. Previous research in other regions such as the Ponderosa pine and the mixed conifer landscapes of the Sand Juan Mountains in Colorado show the impact of forest fire on the forest structure [75]. Other regions shown this behavior of ecosystem adaption to forest fires in the Riau peatlands in Indonesia [76]. The urban interfaces are the key problem here [77], and grazing can be the best manager of the forest

ecosystems in the Mediterranean where urban and periurban interfaces with forest, scrublands, and grasslands are a hot spot [78].

## 5. Conclusions

The first PF was set in 2002 and was aimed primarily at controlling shrubland encroachment and allowing the entry of livestock on the TVA plot. Indeed, the annual grazing of goats served to control vegetation regrowth. A second PF was set in 2011 to regenerate the plot's herbaceous species and to enhance plant palatability. The short-term effects of burning were clearly detected in the immediate aftermath of the first PF but were not visible after the second, essentially because of the time allowed to elapse between this fire management episode and the sampling (four years). The results enable us to conclude that three years after the first PF, its effects were no longer perceptible in the chemical properties analyzed. Recurrent fire episodes appear not to have affected the long-term soil properties of this plot, especially when PF applications are more than four years apart. Grazing did not have a major impact on the plot's soil properties because goats only graze the area every six months, which appears to be sufficient time to allow for plant regrowth and for soil properties to recover. In conclusion, the combined land management practices of prescribed fire and livestock grazing did not drastically affect the soil chemical properties, while enhanced soil fertility helped boost the ecosystem's resilience.

**Author Contributions:** Conceptualization, M.A., X.Ú. and A.C.; methodology, M.A., X.Ú. and A.C.; software, M.A., X.Ú. and A.C.; validation, M.A., X.Ú. and A.C.; formal analysis, M.A., X.Ú. and A.C.; investigation, M.A., X.Ú. and A.C.; resources, M.A., X.Ú. and A.C.; data curation, M.A., X.Ú. and A.C.; writing—original draft preparation, M.A., X.Ú. and A.C.; writing—review and editing, M.A., X.Ú. and A.C.; visualization, M.A., X.Ú. and A.C.; supervision, X.Ú. and A.C.; project administration, X.Ú.; funding acquisition, X.Ú. and A.C. All authors have read and agreed to the published version of the manuscript.

**Funding:** CGL2013-47862-C2-1-R and POSTFIRE_CARE Project (CGL2016-75178-C2-2-R [AEI/FEDER, UE]) sponsored by the Spanish Ministry of Economy and Competitiveness and the European Union via European Funding for Regional Development (FEDER). We also thank the FPU Program (FPU13/00139) promoted by the Ministry of Economy, Culture and Sports. We also enjoyed the benefits of grant 2017SGR1344 awarded by the *Agència de Gestió d'Ajuts Universitaris i de Recerca de la Generalitat de Catalunya*, which served to support the activities of the research groups (SGR2017-2019).

**Acknowledgments:** We thank the members of the GRAF team for providing support in the field and helping in completing the project. Finally, we would like to thank the Scientific and Technological Centers at the University of Barcelona (CCiTUB) for undertaking analyses of soil chemical parameters. We thank the two anonymous referees and the editor for the contribution to the final manuscript.

**Conflicts of Interest:** The authors declare no conflict of interest.

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
