# Peer review of "A 13-Year Approach to Understand the Effect of Prescribed Fires and Livestock Grazing on Soil Chemical Properties in Tivissa, NE Iberian Peninsula"

_forests, doi:10.3390/f11091013_

Round 1
Reviewer 1 Report
Aug., 17, 2020
Re: Peer review of “A 13-year approach to understanding the effect of prescribed fires . . .”
Overall: Well-written. Good introduction. Conclusions generally supported by data. My major concern is how the authors interpret the combined effects of fire and livestock grazing as will be explained later. A secondary concern centers on methodology.
Introduction:
Line 43, insert as after such.
Line 51, [6, 7]; thereby affecting the fate . . .?
Good use of photos.
L 114, soil nutrient availability and soil erosive potential?
Methods:
L 163-170, Unclear how sampling points determined? Randomly? Given the spatial variability of pre-burn vegetation, one would expect considerable heterogeneity in fire effects. I understand why ash might be removed from samples prior to analyses, but so difficult to quantitatively separate ash from mineral soil. Wouldn’t it also be important to include ash in the analyses? Ash can greatly increase soil nutrient availability post-fire. I am not familiar with the C and N analyzer used. If it uses high temperature oxidation to quantify C and N; then there is a possibility that inorganic C from carbonates will be included with organic C? Was expecting some measure of N availability pre-and post-fire? Fire can greatly affect N availability. In regard to sampling after the fire, was ash removed for samples taken 1, 3, and 13 years after the fire. Seems in that amount of time ash may have blown away or had been incorporated into the mineral soil. Seems like a comparison between initial fire effects and later sampling could be confounded by the reality? My biggest concern regarding methodology is livestock grazing. Yes, it is acceptable that the results represent the combined factors of prescribed burning and livestock. However, the authors really have not determined the particular role of grazing. A more thorough experimental design would have include exclusion plots. Certainly without livestock grazing the vegetation composition with different and potential the soil chemistry as well. Ammonium acetate extraction measures extractable base cations, which is a combination of exchangeable plus soil solution-phase base cations.
Results:
Table 1, My copy does not bold mean?
Table 1, ES should be SE?
Table 2, Nothing bolded?
Table 2, Shouldn’t the p-values have four places? What does 0.000 equal?
I appreciate that base cations are affected by fire and vegetation, but in soils developed in calcareous parent material, do even large changes in extractable calcium matter?
I did not see Table 3 references in the paper? I would delete as it adds nothing to paper.
Discussion:
The first paragraph discusses the use of livestock grazing to reduce fuel loads and reduce fuels particularly shrub vegetation. While certainly a laudable goal, no data is provided on how goat grazing affects vegetation composition. The experimental design simply does not allow any inferences regarding grazing activities. If the papers purpose is the quantify fire/grazing effect on soil properties, should not the discussion lead with this?
L 285, How can authors state that pH and EC levels are not affected by grazing when there has been no grazing control?
L 308, Authors cite references relating to how incorporation of ash can increase N stocks; yet, they apparently excluded ash in at least some of their measurements?
L 316-325, Again, to definitively discern the affects of livestock grazing on soil properties, need to have ungrazed control plots.
Section 4.3, Soil quality is quite a nebulous concept. While C and N stocks can certainly relate to soil quality, some measure of soil biological activity after the prescribed fire and livestock grazing would be more informative as to soil quality status. The use of prescribed fire to lower the risk of catastrophic wildfire coupled with livestock grazing to reduce encroachment of undesirable and more fire prone vegetation is a laudable goal in itself; a sustainable system to support humans.
Author Response
Reviewer 1. Forest, Alcañiz et al., 2020
Aug., 17, 2020
Re: Peer review of “A 13-year approach to understanding the effect of prescribed fires . . .”
Overall: Well-written. Good introduction. Conclusions generally supported by data. My major concern is how the authors interpret the combined effects of fire and livestock grazing as will be explained later. A secondary concern centers on methodology.
Thanks
Excellent review and very helpful comments
Introduction:
Line 43, insert as after such.
Done
Line 51, [6, 7]; thereby affecting the fate . . .?
We introduced Infiltration
The sentence will be now
And fire affects the fate of soil infiltration, soil erosion processes and rates, vegetation changes, soil properties and ecosystems services [8-12]
Good use of photos.
Thanks, we think this is important for the reader
L 114, soil nutrient availability and soil erosive potential?
This comment was added
Methods:
L 163-170, Unclear how sampling points determined? Randomly?
Yes, we added this information based on a previous publications. The sampling was based on a regular grid
We try to avoid to repeat this information as it is in
Alcañiz, M.; Outeiro, L.; Francos, M.; Farguell, J.; Úbeda, X. Long-term dynamics of soil chemical properties after a prescribed fire in a Mediterranean forest (Montgrí Massif, Catalonia, Spain). Sci. Total Environ. 2016, 572, 1329-1335.
Given the spatial variability of pre-burn vegetation, one would expect considerable heterogeneity in fire effects.
Right, this is the usual finding
I understand why ash might be removed from samples prior to analyses, but so difficult to quantitatively separate ash from mineral soil.
Right, this was done carefully in each sample
Wouldn’t it also be important to include ash in the analyses?
Other papers did this before in our research team and in the international scientific community
Ash can greatly increase soil nutrient availability post-fire.
Yes, this is right
I am not familiar with the C and N analyzer used. If it uses high temperature oxidation to quantify C and N; then there is a possibility that inorganic C from carbonates will be included with organic C?
Your thoughts are right but in our case this did not happed due to the temperature used
Was expecting some measure of N availability pre-and post-fire? Fire can greatly affect N availability.
Right, but did not take place in our study site
In regard to sampling after the fire, was ash removed for samples taken 1, 3, and 13 years after the fire.
Yes, was done
Seems in that amount of time ash may have blown away or had been incorporated into the mineral soil. Seems like a comparison between initial fire effects and later sampling could be confounded by the reality?
Yes, but this is what we found in nature. And many of the ash blow or wash away
My biggest concern regarding methodology is livestock grazing. Yes, it is acceptable that the results represent the combined factors of prescribed burning and livestock. However, the authors really have not determined the particular role of grazing.
We understand your concern. We analyzed a combined effect and not the separated effects as our objective is to analyze the current management policies with goats and prescribed fires, not the grazing alone or the prescribed fires alone.
But your suggestion is very interesting as this should be the next step in our research
A more thorough experimental design would have include exclusion plots. Certainly without livestock grazing the vegetation composition with different and potential the soil chemistry as well. Ammonium acetate extraction measures extractable base cations, which is a combination of exchangeable plus soil solution-phase base cations.
This is a very interesting idea we should develop in the near future in our team
Results:
Table 1, My copy does not bold mean?
Done
Table 1, ES should be SE?
Done
Table 2, Nothing bolded?
Done. This is a very relevant update
Table 2, Shouldn’t the p-values have four places? What does 0.000 equal?
We checked the Statistics and updated and is correct
I appreciate that base cations are affected by fire and vegetation, but in soils developed in calcareous parent material, do even large changes in extractable calcium matter?
No, at least not in our study site in Tivissa
I did not see Table 3 references in the paper? I would delete as it adds nothing to paper.
They are in the first column as citations and listed in the reference list
Table 3. Mean chemical property values recorded at the TVA plot on the last sampling date (13 years after first PF) compared with values reported in soil reports for the area. B, Basic; H, High; M, Medium; L, Low; U, Up optimal)
|
pH |
EC |
TC |
SOC |
TN % |
Ca2+ ppm |
Mg2+ |
K+ |
Ava P ppm |
|
|
µS cm-1 |
% |
% |
|
|
ppm |
ppm |
|
Tivissa 13y |
7.75 |
167.89 |
8.78 |
5.37 |
0.58 |
17091.73 |
262.07 |
557.91 |
77.98 |
[66] |
>6.5 B |
- |
- |
- |
- |
>3000 H |
>245 H |
>25 H |
>30 H |
[64] |
>7 B |
<500 L |
- |
>2 H |
- |
>400 H |
>30 H |
>100 U |
>50 U |
[63] |
- |
100-450 L |
- |
- |
- |
>2000 H |
>180 H |
280-800 H |
40-100 H |
[65] |
7-8 M |
100-400 L |
- |
>3.5 H |
>0.2 H |
>4000 H |
<300 L |
>300 H |
>30 H |
[67] |
- |
- |
- |
- |
- |
>300 H |
>100 H |
>80 H |
>8 H |
[68] |
7-8 M |
- |
- |
4-10 M |
>0.5 H |
>2000 H |
60-500 M |
>250 H |
>15 H |
[62] |
7.5-8.5 B |
<400 L |
- |
- |
- |
- |
- |
- |
- |
Discussion:
The first paragraph discusses the use of livestock grazing to reduce fuel loads and reduce fuels particularly shrub vegetation. While certainly a laudable goal, no data is provided on how goat grazing affects vegetation composition. The experimental design simply does not allow any inferences regarding grazing activities. If the papers purpose is the quantify fire/grazing effect on soil properties, should not the discussion lead with this?
Thanks for this advice
We discuss the general impact of grazing under the light of the literature, not based only in our data
Our discussion is initiated as
“Previous researches have shown that the effects of burning….”
L 285, How can authors state that pH and EC levels are not affected by grazing when there has been no grazing control?
Thanks fo the comment
We discuss the general impact of grazing under the light of the literature, not based only in our data
Our section on this topic was initiated as follow “A good many study conducted in different ecosystems around the world show that soil pH increases immediately after a PF [29”
L 308, Authors cite references relating to how incorporation of ash can increase N stocks; yet, they apparently excluded ash in at least some of their measurements?
Yes, this is right, this is a short note to mention that ash can be also a relevant topic to be researched in the future as we did in the past
L 316-325, Again, to definitively discern the affects of livestock grazing on soil properties, need to have ungrazed control plots.
Right. We studied a real situation with our survey… not separated (experimentally) situations. Our discussion on the effect of grazing is based in the literature
Section 4.3, Soil quality is quite a nebulous concept. While C and N stocks can certainly relate to soil quality, some measure of soil biological activity after the prescribed fire and livestock grazing would be more informative as to soil quality status. The use of prescribed fire to lower the risk of catastrophic wildfire coupled with livestock grazing to reduce encroachment of undesirable and more fire prone vegetation is a laudable goal in itself; a sustainable system to support humans.
Right. We fully agree with your opinion, and this is why we developed this research

Reviewer 2 Report
This is a relatively well-written manuscript describing an integrated approach of fire and livestock grazing management on soil properties of forested lands. Specific comments are:
- Page 1, line 25: Delete “that used to be”.
- Page 1, line 29: Change “boost” to “boosted”.
- Page 1, line 38: Change “…along the Quaternary” to “during the Quaternary Period”.
- Page 1, line 39: Insert “Thus” at the beginning of the sentence before “the”.
- Page 2, line 49: Insert “natural” bet ween “the “ and “Mediterranean”.
- Page 5, lines 168-169: The references for the analytical methods seem to be switched. Please check to see if they are correct.
- Page 5, Table 1: The authors need to check the abbreviations in the table heading and also within the table: especially ES and SE.
- Page 6, line 200: Change “Along” to “During” at the beginning of the sentence.
- Page 6, lines 202-207: What were the changes in inorganic C (IC)? If IC estimated by the difference in TC-SOC, IC increases about 4-fold from pre-fire to 1 year after and its presence continues throughput the 13-year time period. This appears to be confirmed by the pH values. This appears to be a significant change in soil chemical properties that may impact soil erosion and even growth of some plant species. This should be addressed as one of the soil properties. Inorganic C can also be a form of C conservation in the soil system. The IC may also be related in the decrease in Ca2+ after PF.
- Page 8, line 246: Change “researches” to “studies”.
- Page 8, line 271: Change “study” to “studies”.
- Page 10, line 373: Insert “after” between “that” and “a”.
- Page 10, line 375: Change “atmosphere” to “atmospheric”.
- Page 10, line 377: Change “is” to “in”; change “area” to “areas”.
- Page 10, lines 382-386: This section needs careful editing to make it more readable. English editing?
- General comment: The authors need to evaluate the use of too many digits when reporting numeric values. Use the 3 significant digit rule.
This manuscript should be accepted for publication after the authors address the above concerns. Minor English usage and grammar editing should also be addressed.
Author Response
Reviewer 2. Forest, Alcañiz et al., 2020
Re: Peer review of “A 13-year approach to understanding the effect of prescribed fires . . .”
This is a relatively well-written manuscript describing an integrated approach of fire and livestock grazing management on soil properties of forested lands.
Thanks for the comments and the contribution to improve the paper
Specific comments are:
- Page 1, line 25: Delete “that used to be”.
Done
- Page 1, line 29: Change “boost” to “boosted”.
Done
Page 1, line 38: Change “…along the Quaternary” to “during the Quaternary Period”. Done
Page 1, line 39: Insert “Thus” at the beginning of the sentence before “the”. Done
Page 2, line 49: Insert “natural” bet ween “the “ and “Mediterranean”. Done
Page 5, lines 168-169: The references for the analytical methods seem to be switched. Please check to see if they are correct. Done
Soil samples were taken at each sampling point from the top layer (0-5 cm) using a small pick. Ash was then removed and the sample air dried in the laboratory and sieved to obtain a <2 mm fraction. Following extraction with deionized water, soil pH (1:2.5) and Electrical Conductivity (1:2.5) were analyzed and measured with a pH-meter and a conductometer, respectively [43]. Total C and total N were determined using elemental analysis (NaA2100 Protein Nitrogen Analyzer). Available P was analyzed using the Olsen Gray method [44]. Exchangeable cations, K+, Ca2+ and Mg2+ were analyzed by ammonium acetate extraction [45] and determined by atomic absorption spectrophotometry.
44.Olsen, S.R.; Cole, C.V.; Frank, S.W.; Dean, L.A. Estimation of available phosphorus insoils by extraction with sodium bicarbonate. 1954. USDA Circular No. 939. US Government Printing Office, Washington, DC.
45.Knudsen, D.; Petersen, G.A.; PF, P. Lithium. Sodium and potassium. In: Soil Science Society of America (Ed.). 1986. Methods of soil analysis vol.2. ASA-SSSA, Madison, pp. 225–246.
Page 5, Table 1: The authors need to check the abbreviations in the table heading and also within the table: especially ES and SE. Done
Page 6, line 200: Change “Along” to “During” at the beginning of the sentence. Done
Page 6, lines 202-207: What were the changes in inorganic C (IC)? If IC estimated by the difference in TC-SOC, IC increases about 4-fold from pre-fire to 1 year after and its presence continues throughput the 13-year time period. This appears to be confirmed by the pH values. This appears to be a significant change in soil chemical properties that may impact soil erosion and even growth of some plant species. This should be addressed as one of the soil properties. Inorganic C can also be a form of C conservation in the soil system. The IC may also be related in the decrease in Ca2+ after PF. Done, We introduced this interesting view in our results
Page 8, line 246: Change “researches” to “studies”. Done
Page 8, line 271: Change “study” to “studies”. Done
Page 10, line 373: Insert “after” between “that” and “a”. Done
Page 10, line 375: Change “atmosphere” to “atmospheric”. Done
Page 10, line 377: Change “is” to “in”; change “area” to “areas”. Done
Page 10, lines 382-386: This section needs careful editing to make it more readable. English editing? Done. This was a weak writing and now is more consistent
General comment: The authors need to evaluate the use of too many digits when reporting numeric values. Use the 3 significant digit rule. Done
This manuscript should be accepted for publication after the authors address the above concerns. Minor English usage and grammar editing should also be addressed.
